# 'You have got a foreign body in there': renal transplantation, unexpected mild-to-moderate distress and patients' support needs: a qualitative study

Janet Jones  ,[1] Sarah Damery,[1] Kerry Allen,[2] Johann Nicholas,[3] Jyoti Baharani,[4] Gill Combes[1]

¹Institute of Applied Health Research, University of Birmingham, Birmingham, UK
²Health Services Management Centre, University of Birmingham, Birmingham, UK
³Renal Unit, Shrewsbury and Telford Hospital NHS Trust, Shrewsbury, UK
⁴Renal Unit, Birmingham Heartlands Hospital, University Hospital Birmingham NHS Foundation Trust, Birmingham, UK

**Correspondence to**
Dr Janet Jones;
j.e.jones@bham.ac.uk

## ABSTRACT

**Objective** To explore why transplant patients experience unexpected mild-to-moderate distress and what support they may need.

**Design** Qualitative study using individual in-depth interviews.

**Setting** Four National Health Service (NHS) Trusts in the Midlands, UK.

**Participants** Fifteen renal transplant patients meeting the criteria for mild-to-moderate distress from their responses to emotion thermometers.

**Main outcome measures** Identification of the reasons for distress and support options acceptable to renal transplant patients.

**Results** Three themes were interpreted from the data: 'I am living with a "foreign body" inside me', 'why am I distressed?' and 'different patients want different support'. Following their transplant, participants felt that they should be happy and content, but this was often not the case. They described a range of feelings about their transplant, such as uncertainty about the lifespan of their new kidney, fear of transplant failure or fear of the donor having health conditions that may transfer to them. A few experienced survivors' guilt when others they had met at the dialysis unit had not received a transplant or because someone had died to enable them to receive the transplant. No longer having regular contact with the renal unit made participants feel isolated. Some participants did not initially attribute the source of their distress to their transplant. Participants' preferred support for their distress and their preferences about who should deliver it varied from peer support to seeing a psychologist.

**Conclusions** Raising the issue of post-transplant mild-to-moderate distress with patients and encouraging them to think about and plan coping strategies pretransplant may prove beneficial for the patient and healthcare provider. Patients should be able to choose from a variety of support options.

## INTRODUCTION

At the end of 2017, 63 162 adults were receiving renal replacement therapy in the UK,[1] and in the same year, 3462 transplants were performed.[1] Patients with end-stage renal disease (ESRD) can experience distress

### Strengths and limitations of this study

► This is the first study designed to explore the issue of unexpected mild-to-moderate distress in renal transplant patients.
► The interviews were a subset of data from a larger study.
► To reduce bias on the basis of the services offered to renal patients multiple sites with different organisations and delivery of services were selected.
► Participants were diverse thus the findings are more representative of the wider group of renal transplant patients.

and distress is associated with lower quality of life and greater treatment burden.[2] The level of emotional distress increases as the health of a patient with ESRD declines[3] and the prevalence of depression and anxiety in patients with ESRD is approximately four times higher than in the general adult population.[4] Damery *et al*[5] reported that more than a third of renal dialysis patients suffer emotional distress. If suffering from mild-to-moderate distress—the unpleasant feelings or emotions that may interfere with patients' ability to cope with their kidney transplant, its physical symptoms and its treatments—patients may withdraw from treatment, be non-compliant with medication and diet[6] or be reluctant to engage in prerenal replacement therapy (RRT) education and support.[7]

Although not a cure, for patients with ESRD, receiving a transplant provides the best chance of improving their quality of life and removing the burden of undergoing long-term dialysis treatment.[8 9] Nevertheless, research has shown that although receiving a transplant is the main goal for patients with ESRD who are eligible for transplantation, they may continue to experience distress. For example, fear of the transplant failing can

affect patient distress levels.[8] There is also evidence that patients may experience ongoing physical symptoms such as fatigue after receiving a transplant, and that there is a substantial burden associated with taking regular immunosuppressant medication to lower the likelihood of graft rejection.[10] UK health policies highlight the importance of addressing the emotional and psychological needs of renal transplant patients[11] and the Department of Health and NHS England advocate treating mental health on an equal footing to that of physical health by incorporating it into care pathways.[12] Nevertheless, evidence suggests that distress, coping and adjustment in transplant patients largely go undiagnosed or ignored and remain untreated.[13] Currently, there is little evidence on mild-to-moderate distress in transplant patients and further research is required in order to understand the psychological and emotional effects of a transplant.[9 14 15]

As an element of a larger mixed-methods study with patients and staff,[16] this paper reports the findings from a qualitative study with renal transplant patients. The aims were to explore why transplant patients experience distress and what support they may need.

## METHODS

The detailed methods for the study are in the published protocol.[16] In brief, participants were recruited from four National Health Service (NHS) hospitals in the West Midlands, UK. The chosen sites provided maximum diversity in patient demographics, catchment size, urban and rural locations plus the organisation of psychological support services and were categorised according to the size of the catchment area (table 1).

To be eligible for recruitment to the qualitative study, patients had to be categorised as having mild-to-moderate distress based on their responses to the distress thermometer[17] included in the wider study questionnaire. Participants indicated on the questionnaire whether or not they would be willing to take part in an interview and provided their preferred contact details. A consent form and participant information sheet explaining the purpose of the interview study and what participation would involve were sent to those expressing interest. The following week those meeting the purposive sampling criteria (age, sex and ethnicity) were contacted by a researcher (FT) to confirm participation and arrange a date and time for the

interview. If participants had changed their mind, they were able to withdraw from this part of the wider study. As well as assuring patient confidentiality, the patient information sheet provided contact details of appropriate clinical staff the participants could contact if they felt distressed or upset and would like support. Contacting the participants to answer their questions, to arrange the interview and to remind them a few days beforehand helped to build up a rapport with each participant prior to their interview.

Patient interviews took place between March 2016 and May 2017 and were conducted by two experienced qualitative researchers (FT, EK) (both identifying as female and educated to masters level) employed by the University of Birmingham. Neither researcher had experience (personally or professionally) of the topic area and none of the participants knew the researchers. Interviews were either face-to-face (at a location chosen by the participant, either a quiet room at the hospital or at the patient's home) or over the telephone and lasted between 30 and 60 min. The interviews were in-depth and semi-structured allowing the exploration of key issues without being overly prescriptive about content and direction. All participants provided signed written informed consent prior to their interview. Patient advisors, renal clinicians and the current literature helped with the design of the topic guide (see box 1 for the areas covered by the topic guide). Prior to the start of the interview, participants were advised to let the researcher know if they needed to take a break during the interview or if they no longer wished to carry on. Although included in the participant information sheet, participants were reminded that taking part in an interview would not impact the care they receive. At the end of each interview, participants were asked if they had any further comments on the topics covered or whether there were any important areas they felt had not been discussed. Both researchers made field notes following each interview. Interviews were audio-recorded and transcribed verbatim by a professional transcribing service. Transcripts were checked against the recordings for accuracy. Participants did not have the opportunity to review their interview transcripts.

Analysis combined aspects of grounded theory[18] and thematic analysis.[19] Interviews were initially analysed inductively using the open coding and constant

| Table 1 | Summary of recruitment sites | | |
|---------|------|------|------|
| Site | Size | Catchment area | On-site renal psychologist |
| 1 | Small | Urban inner city with sizeable BAME population | No |
| 2 | Large | Urban inner city with sizeable BAME population | Yes |
| 3 | Medium | Urban with surrounding rural districts majority white population | No |
| 4 | Large | Urban with surrounding rural districts majority white population | Yes |

BAME, Black, Asian and minority ethnic.

## Box 1 Areas covered by the topic guide.

- ► Experience of emotional difficulties and needs linked to their illness and/or treatment, when and for how long.
- ► Language used around emotional difficulties and needs, and its meaning.
- ► Whether and how emotional needs have been recognised and supported by renal staff, when and by whom.
- ► What, if any, support used, when and why.
- ► Likes and dislikes of support used.
- ► Support patients would have liked/would want in future, when and from whom.
- ► Key elements patients would like included in an emotional support intervention.

comparison aspects of grounded theory. The initial coding framework was developed by JJ and GC and was appropriately refined following comparison and discussion. Transcripts were coded using NVivo 11. For data that did not fit existing themes, new codes were developed or existing ones revised until all data were coded by theme. Following the completion of data collection, the research team met and agreed that in order to understand participants' experiences and to help inform future practice, a generic pragmatic hybrid approach to analysis was appropriate.[20][21] The research team also discussed the role of reflexivity and how our personal views and experiences may influence our interpretation of the data.[22][23]

### Patient and public involvement

The patient and public involvement group of the NIHR CLAHRC West Midlands long-term conditions theme and a renal patient advisory group provided advice on the design of the study, the data collection tools and the selection of outcomes. All participants received a summary of the study findings.

### RESULTS

Fifteen renal transplant patients aged between 30 and ≥70 years were recruited across the four sites (tables 2 and 3).

Three overarching themes were interpreted from the data. See table 4 for a summary of the themes and subthemes.

### I am living with a 'foreign body' inside me
#### Fear/feelings about the kidney itself

Living with a transplanted kidney and its associated treatments can evoke many different feelings and difficulties for patients. For many, there is the fear and uncertainty of how long the transplant will last: participants talked about their transplant having a finite life and how this knowledge made them worry about what would happen in the future. Three participants had experienced a failed transplant and others understood the status of their transplant could change at any time:

A lot of people I knew before when I was on dialysis who had transplants, they had rejections and all sorts

of things. So that's passing through your mind all the while. (P225).

Among some patients, their fear had escalated the longer they had been with a transplant, particularly

### Table 2 Summary of participant characteristics

| | N (%) |
|---|---|
| **Sex (participant self-identified)** | |
| Male | 7 (47) |
| Female | 8 (53) |
| **Age (years)** | |
| 30–39 | 3 (20) |
| 40–49 | 2 (13) |
| 50–59 | 4 (27) |
| 60–69 | 4 (27) |
| ≥70 | 2 (13) |
| **Ethnicity (participant self-identified)*** | |
| White | 9 (60) |
| Indian | 4 (27) |
| Caribbean | 2 (13) |
| **Length of time on dialysis prior to transplant** | |
| No dialysis | 1 (7) |
| 0–3 years | 5 (33) |
| 4–9 years | 0 (0) |
| ≥10 years | 3 (20) |
| Unknown | 6 (40) |

*According to the Office of National Statistics ethnicity groupings 2015.

### Table 3 Individual participant characteristics

| ID | Age (years) | Time on dialysis prior to transplant |
|---|---|---|
| P129 | 40–49 | No dialysis |
| P267 | 60–69 | Not stated |
| P384 | 50–59 | 16½ years |
| P413 | 30–39 | Not stated |
| P494 | 50–59 | 11 years |
| P687 | 30–39 | Not stated |
| P726 | 50–59 | 10 years |
| P781 | ≥70 | 11 months |
| P197 | 40–49 | 1 year |
| P225 | ≤70 | 3 years |
| P369 | 60–69 | Not stated |
| P389 | 50–59 | 3 years |
| P401 | 30–39 | Not stated |
| P1028 | 60–69 | 6 months |
| P1141 | 60–69 | Not stated |

**Table 4** Themes and subthemes

| Theme | Sub-themes |
|---|---|
| I am living with a 'foreign body' inside me | Fear/feelings about the kidney itself |
| | Survivors' guilt |
| | Feelings of isolation |
| | Impacts of medication |
| Why am I distressed? | Expectations of living with a transplant |
| | Coping with distress |
| | Lack of information about transplants and support |
| Different patients want different types of support | |

after their transplant exceeded the average lifespan. Some patients thought the longevity of a transplanted kidney was something of a lottery, which was perceived to be outside of their control and could therefore cause distress. A few patients seemed emotionally affected by feeling they had a 'foreign body' inside them (M225). One male patient talked about the possible consequences for his character of having received a female kidney. Another was anxious that his transplanted kidney might have come from someone with other health problems:

> When I got it I started to get awful thoughts at night. 'How did this person die, how old were they, did they have anything else, could they possibly have been HIV positive? Could there have been any other things that were underlying that may come forward later on?' and I still get those sort of thoughts at the moment. (P1141).

### Survivors' guilt

Although a transplant is the gold-standard treatment for ESRD, some participants mentioned feeling guilty about being distressed and some worried that this may make them appear ungrateful. They felt that they should be happy because they were lucky to receive a transplant: *'but you, at the same time you've got no reason to feel like that'*. (M389). This guilt and the fear of appearing ungrateful prevented some from seeking help when they needed it and stopped them from moving forward with their lives. Some participants worried about the donor, what happened to them and the family they left behind and others had survivors' guilt because they had received their transplant before others who had been waiting longer:

> When I had my transplant I felt, I suppose what you'd call it is like in a sense survivors guilt…And because I had my transplant so quickly I just felt this huge amount of guilt because I thought all those patients that I had met at (hospital) and they've been on the list for such a long time, and I thought 'what's the difference between me and them?'. (P413).

### Feelings of isolation

There was a perception among several of the participants that after receiving a transplant, they were no longer part of the renal unit. They talked about feeling 'cut-off' and 'abandoned' with minimal contact. One patient explained how there had been no contact with his renal unit for several years following his transplant. Patients who had transitioned from in-centre haemodialysis (HD) seemed to feel this loss acutely—they missed the supportive relationship of staff and patients in the dialysis unit:

> Once you're transplanted, you don't really have anything to do as such with the renal unit……the only time I actually only ever was involved with the renal unit was when I was on dialysis. So that sort of like support was gone. (P726).

Some patients experienced feelings of isolation because of the lack of understanding about their disease and its treatments among family, friends and society in general. They believed only fellow transplant patients understood their feelings:

> It helped the fact talking to a stranger about it. You know, the wife couldn't understand why I was like it and that, but obviously the stranger could because she knew the experience of it all. (P369).

### Impacts of medication

Feeling down and finding their situation difficult to deal with was widespread among the participants. These feelings were often present when participants were dealing with the physical side effects of the anti-rejection medications such as weight gain, puffy appearance and excessive hair growth. Other peoples' reactions to the physical changes were often difficult to deal with:

> You know like the new face and excessive hair … and I found that I think the most difficult for me was that change and unfortunately when I went back to school I had a bit of, a bit of bullying went on, because of physically changing so much from what I was previously as well. (P687).

Participants explained how when they were on dialysis, they felt in control of their body, their treatment and their lives but following transplant, many felt they no longer had command of their body or life in general and yearned to take back control. Such as, not wanting to leave the house because of the lack of bladder control. For the following patient, the feeling of helplessness ended with her sabotaging her treatment:

> So the reason my kidney failed was because I felt like I didn't have much control, so I kind of stopped taking my pills, my immunosuppressant. I stopped taking them for a while. (P413).

## Why am I distressed?
### Expectations of living with a transplant

Following their transplant, participants did not expect to experience distress because they regarded a transplant as the best treatment option for an improved quality of life. Some patients had waited many years for the opportunity to receive the 'gift' of a transplant and had been optimistic:

> I'm coming at it first time with an expectation that once you get a transplant and you start feeling better again, life is rosy, life can get back to normal. (P389).

While experiencing distress, many participants did not initially relate it to their transplant. The data suggest that the reasons for this were twofold. First, following a transplant, there is an expectation from family, friends, clinicians and society in general that life will return to 'normal' and be 'wonderful' again.

> I never did link it to – there was something in your paperwork that I had. I thought I've never associated it with the kidney operation. (P369).

Second, renal staff did not forewarn the participants that they might experience distress after receiving a transplant. Some only made the connection after agreeing to take part in this research study:

> But being in the hospital and then having all sorts of side effects which is effecting me emotionally now. And that things could have been told to me before, you know, I would have prepped myself up. Some of those things could have been addressed before. It could have been helpful to have been told beforehand. (P129).

### Coping with distress

Renal disease and its life cycle can make it hard for patients to be positive about their transplant. Several participants explained how it was difficult to cope with their emotions, to move on with their lives and how they perceived family and friends, in particular, were defining them by their health condition:

> (Sighs) Ah, again I suppose it comes back to the fact that I don't want to like, yes like I know I've got an illness, but I don't want to be that's who I am like. You know that's what I'm all about sort of thing. (P726).

The majority of participants developed ways of accepting post-transplant life and found ways to cope with their distress. For some, distractions in the form of hobbies and pastimes such as gardening or reading helped them to adjust and for others, it was important to maintain a positive outlook about their transplant:

> I've always had this positive outlook on the transplant and that. (P225).

### Lack of information about transplants and support

Many participants wished the renal team had explained to them the possibility of experiencing post-transplant distress and lamented the lack of information about this and the lack of available support. Many felt that 'forewarned is forearmed' and were upset they missed the opportunity to plan, in advance, coping strategies. This lack of information provision was associated with a lack of continuity of care, and participants felt that staff regarded transplant patients as a lower priority compared with those on dialysis:

> Time is precious, resources are scarce, I suspect that probably it never feels as though it's a big priority. (P389).

### Different patients want different types of support

There were, however, diverse views among the participants about the types of support they would like to receive, and who should deliver it. For example, one participant described how she thought talking to other patients might have provided the support she sought.

> So with other patients in your situation, so that you could liaise with each other to see what new life is all about. (P494).

There were mixed views about the role of healthcare professionals (HCPs), with some suggesting that HCPs should be more proactive in identifying distress in renal transplant patients and directing them to appropriate sources of help and support. Whereas others did not see this as the role of the renal team—they are there to provide medical assistance and advice not psychological and emotional support. Many suggested that specialist psychological services should be available as an integral part of care for renal transplant patients:

> I think you need psychologists as part of the renal team, a psychologist with renal expertise. (P781).

However, because of the sensitive and personal nature of distress or for fear of showing weakness, some were reluctant to talk about their distress to anyone, making it difficult for HCPs to assess the support needs of these patients:

> You know, it might have been something to do with medication – I'm just guessing. But I should have told them but as I say, I mean it's not the sort of thing I tell people (P369).

## DISCUSSION

This research has highlighted the complex relationship that renal transplant patients have with their new kidney, their largely unanticipated experience of their distress and the diverse opinions on the types of support they would like to help them through their distress. Current research has shown that the prevalence of distress in

renal transplant patients is 25% and although this is lower than the 33% of dialysis patients, it is still substantial and shows that many transplant patients experience ongoing issues.[5] There is a range of reasons why despite all expectations of living an improved life, some patients became uneasy with their transplanted kidney and why some did not associate their distress with their transplant and/or immunosuppressant medication.

Patients found it difficult to accept their new kidney because of a number of different situations including: the side effects of medication, fear of transplant rejection and feeling obliged to make the most of life because of the 'gift' of a transplant.[24–27] It has been reported that concern about the potential lifespan of the transplanted kidney may be the biggest stressor immediately after transplant, this fear subsides with time.[8] However, our findings do not reflect this: the majority of our participants, regardless of time since transplant, reported some level of emotional distress.[5] Emotional problems such as depression, anxiety, stress and concerns about body image are known indicators of poor medication adherence[3 6] and as discussed, it may result in patients wishing they were back on dialysis and in some cases sabotaging their treatment. Gill,[8] Suzuki et al[28] and Fox[29] found that those receiving a kidney from a living donor had a vested interest in each other's well-being and were motivated to comply with their medication regimes and enjoyed an improved quality of life. For those receiving a cadaver kidney, thoughts about who their kidney came from were forefront in their mind and for some this had a negative effect on their relationship with their kidney—making them feel as though they had a 'foreign body' inside them. The emotional distress patients feel may be down to unrealistically high expectations of life post-transplant and there is a need for these expectations to be managed by renal services.[8 25 30]

The majority of participants did not expect to experience distress after a transplant. This may be that patients naively perceive that by having a transplant their quality of life will improve quickly and they will return to a normal life.[31] Consequently, patients are often ill-prepared and feel helpless when trying to cope with their distress.[32] Evidence has shown that improving coping skills,[33] education before and after transplantation,[34] and active information seeking by patients can have a beneficial effect on patients' medical and psychological problems.[8 25 35] The National Institute for Health and Care Excellence (NICE) advocate keeping patients informed at all stages of treatment and encourage the promotion of self-care and self-management skills.[36] However, HCPs can find it difficult to recognise distress or anxiety in patients and are unsure at which point in the disease trajectory to discuss this with their patients.[37] When a patient is informed, it empowers them to take control of their condition and having control can itself lower the chances of distress.[6] It is therefore important that HCPs provide and share information and discuss all possible outcomes and coping strategies at the appropriate time in the treatment pathway.[3 30 32]

Not all renal transplant patients will experience distress and not all of those who do will want to receive support. This research has shown that some patients do not wish to talk about their feelings for fear of appearing ungrateful or weak, making it difficult for HCPs to understand, assess and procure appropriate support.[37] Even when patients do want to talk about their feelings, there is 'no one size fits all' solution to the provision of support.[38] Different patients have different emotional needs indicating that any support offered to patients should be individualised in order to meet this variance of need.[39]

There were no discernible differences in the depth of data and the length of the interview between those conducted face-to-face and those conducted over the telephone. A limitation of this study is that the interview data analysed for this research is a subset of a larger set of data. Although the 15 in-depth interviews provided sufficient data to answer the research questions and were more representative of the wider group of transplant patients, future research needs to explore these areas with a more diverse and carefully stratified sample.

## CONCLUSIONS

Our research has highlighted a number of points: first, it is important to talk to patients and their families pretransplant about the possibility of experiencing mild-to-moderate post-transplant distress. Second, patients should be encouraged to think about potential coping strategies and finally, transplant patients with mild-to-moderate distress should be able to choose from a variety of support options: peer support; HCPs with augmented skills in detecting and managing distress; access to psychology services.

**Acknowledgements** Francesca Taylor and Elaine Kidney conducted the interviews. Pamela Nayyar provided administrative support for this research. The patient and public involvement group of the CLAHRC West Midlands long-term conditions theme and a renal patient advisory group for their invaluable help and advice. Finally, a huge thanks goes out to all of the participants who gave up their time to talk to the authors.

**Contributors** JJ: conceptualisation (for this sub-study), formal analysis, interpretation of data, methodology, validation, writing (original draft). GC: formal analysis, validation and writing (critical review and editing). SD and KA: interpretation of data and writing (critical review and editing). JN and JB: conceptualisation, validation and writing (critical review and editing). All authors have approved the final version.

**Funding** This research was funded by the National Institute for Health Research Collaboration for Leadership in Applied Health Research and Care West Midlands (NIHR CLAHRC WM), now recommissioned as NIHR Applied Research Collaboration West Midlands.

**Disclaimer** The views expressed in this publication are those of the author(s) and not necessarily those of the NIHR or the Department of Health and Social Care.

**Competing interests** None declared.

**Patient and public involvement** Patients and/or the public were involved in the design, or conduct, or reporting, or dissemination plans of this research. Refer to the Methods section for further details.

**Patient consent for publication** Not required.

**Ethics approval** The Coventry and Warwickshire Research Ethics Committee granted approval for this and the wider study in October 2015 [Ref 15/WM/0288].

The Research Governance office at each of the participating hospital trusts also gave approval for the study.

**Provenance and peer review** Not commissioned; externally peer reviewed.

**Data availability statement** No data are available. The research data are confidential. Participants did not give consent to share their data and the ethical requirements of the study do not allow us to share the study data.

**Open access** This is an open access article distributed in accordance with the Creative Commons Attribution 4.0 Unported (CC BY 4.0) license, which permits others to copy, redistribute, remix, transform and build upon this work for any purpose, provided the original work is properly cited, a link to the licence is given, and indication of whether changes were made. See: https://creativecommons.org/licenses/by/4.0/.

**ORCID iD**
Janet Jones http://orcid.org/0000-0002-9057-6956

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
