## [Reviewer comments · BMJ Open]

ARTICLE DETAILS

TITLE (PROVISIONAL)	"You've got a foreign body in there": Renal transplantation, unexpected mild-to-moderate distress and patients' support needs: a qualitative study
AUTHORS	Jones, Janet; Damery, Sarah; Allen, Kerry; Nicholas, Johann; Baharani, Jyoti; Combes, Gill

VERSION 1 – REVIEW

REVIEWER	Jennifer Poole Ryerson University, Canada
REVIEW RETURNED	03-Dec-2019

GENERAL COMMENTS	Thank you for the opportunity to read this submission and to learn from your research. I quite agree that there is a paucity of qualitative research on the experience of distress for those who receive transplants. I also agree there is much more preparation and support needed for those before and after transplant. Overall, the project has much merit, but for this submission, I would like to see more detail around the theory, methodology and methods. My comments, questions and suggestions for revision are as follows: P.2. Objective: I wonder if the focus of the article is on 'mild-moderate distress' or, as per the title, unexpected distress. Please decide and edit both as necessary for continuity. P.3. Strengths and limitations: "To reduce bias on the basis of the services offered to liver patients multiple sites with different organisation and delivery of services were selected." I realize the study was part of a larger project, but should this read as renal instead of liver patients? P.3. "The diversity of participant characteristics supports the representativeness of the findings to this patient group". What does this mean? Please explain. P.5. "Nevertheless, evidence suggests that distress, coping, and adjustment in transplant patients largely go undiagnosed or ignored and remain untreated.¹³ Currently, there is little evidence on mild-to-moderate distress in transplant patients and further research is required in order to understand the psychological and emotional effects of a transplant. 9." I agree, but there has been more published on distress and quality of life in transplant and specifically in the heart transplant literature. Perhaps that literature would augment this section and better situate the proposed article and its claims.
---

	P.5. Methods: I found reading through the more detailed methods section in the published protocol useful in terms of recruitment (and refusals to participate). P.6. The authors note that potential participants were contacted by one of the researchers. Which one? Was this person also directly involved in the patients' care? Would there be any patient concern about their care if they refused the researcher's request? How was this potential for more distress mitigated as part of the ethics process? P.6. The authors note that both researchers identified as "female". Why was this language used instead of one centering gender (i.e. women-identified)? Was this important to the participants? How? P.7. It is important that the authors note the two interviewers did not previously know the participants. As per protocol around semi-structured interviews however, how was comfort and/or rapport built before the interviews and especially when conducted in the home? How much time was spent with participants prior to the formal interview? What kind of interviewing and/or qualitative research training do the interviewers/researchers have? Were there any differences in the depth and length of interviews between those interviewed in homes as opposed to over the phone? P.7. Were participants shown how to turn off/control the recording if feeling uncomfortable or in distress? P.7. With respect to the interview guide, was any opportunity given to participants to discuss other topics or areas of concern? P.7. The authors note that "Analysis combined aspects of grounded theory (16) and thematic analysis (17). Interviews were initially analysed inductively using the open coding and constant comparison aspects of grounded theory. The initial coding framework was developed by JJ and CG and was appropriately refined following comparison and discussion. Transcripts were coded using NVivo 11." Why did the authors use grounded theory? What was the rationale? Where do the researchers position themselves ontologically and epistemologically vis a vis qualitative research? Were other qualitative methodologies considered and if so what and why? How was the decision made to move from 'thematic analysis' in the first published methods piece to grounded theory in this one? How do the researchers understand reflexivity and methodology? There needs to be more explanation and a clear rationale for theoretical and methodological decisions made by the research team for this project. P.8. Table: Why equate white with British? How were these categories constructed and for what purpose? I found this section quite problematic. P.16 The authors note that "Many suggested that specialist psychological services should be available as an integral part of care for renal transplant patients." Why just psychologists? Did participants discuss or were they made aware of other kinds of support professionals?
--	--

	P.17. How many were surprised by the distress? In some themed areas, you list numbers of participants who spoke to the issue/theme. In others, you do not. Were the majority surprised and if so, please include. P.17. With respect to the gift, there is literature that identifies this as a major pressure for those with transplants. Please cite accordingly. P.18. The authors note, "Given that patients receive little, if any, information about post-transplant distress from HCPs 3 this is not surprising. Consequently, patients are often ill prepared and feel helpless when trying to cope with their distress.2." Given this, it would appear that post renal transplant distress is not so unexpected, at least by the professionals who do not prepare them for it. Please revise this section accordingly. Thank you once again for the opportunity to review this paper.
--	--

REVIEWER	Kenneth Gannon University of East London UK
REVIEW RETURNED	04-Dec-2019

GENERAL COMMENTS	I enjoyed reading this paper and felt that data was interesting and raised some important issues. There are a number of points that would, I think benefit from clarification. Summary: in the third point there is a reference to "liver patients", which I assume is a typographical error. Methods Table 1: Is it possible to briefly describe the basis for the classification of sites by size? Distress thermometer: Is there any information about sensitivity? Results Overall I found the analysis coherent and persuasive and there was some very interesting and important data. However, I thought that there were too many sub-themes that resulted in the analysis being a little lacking in depth at times and that there was scope for developing the interpretative aspects to draw out and integrate what appeared to me particularly significant issues and concerns. For example, I was struck by F413's account of stopping taking medication because of a felt lack of control. This didn't seem to me to fit very well with the sub-theme of the impact of medication and I wondered whether it could be integrated into the sub-theme of coping (coping and control having a close conceptual relationship). It would also be very interesting and, I think, important to understand precisely what sort of control the participant would have wished to have as this might be important for clinicians to understand given the seriousness of the consequences in this instance. I would suggest collapsing the sub-themes "Initial feelings following transplant" and "Association of distress to the transplant" to draw out more clearly what seems to be a central issue of the sorts of hopes and expectations that people have for the surgery and the apparent lack of preparation for the realities. Identifying and addressing unrealistic expectations in advance of surgery has already been identified as important in other areas of surgery,
---

	such as cosmetic and reconstructive procedures. In this respect it would be very helpful to see a quote to support the claim that "renal staff did not forewarn the participants" (P. 14/25). I would also suggest considering bringing the sub-theme "Lack of information about transplants and support" into this theme to pull together everything concerning expectations of surgery and provision of information and guidance prior to surgery. I think that this would also help to distinguish between two types of support that are currently combined but need, I would suggest, to be distinguished. There are support prior to surgery and post-surgical support. Discussion I was not persuaded by the statement that the findings of this study do not support the report that fear of rejection declines with time. Although one participant spoke of fear of rejection it appears from what the authors report that the fear concerned the anticipated lifespan of the transplant rather than rejection per se. NICE is "National Institute for Health and Care (not "Clinical") Excellence".
--	--

VERSION 1 – AUTHOR RESPONSE

Reviewer 1		
	Comment	Reply
	I quite agree that there is a paucity of qualitative research on the experience of distress for those who receive transplants. I also agree there is much more preparation and support needed for those before and after transplant. Overall, the project has much merit	Thank you for your positive view of our manuscript.
P.2.	Objective: I wonder if the focus of the article is on 'mild-moderate distress' or, as per the title, unexpected distress. Please decide and edit both as necessary for continuity.	Thank you for raising this point. The focus of the article is the unexpected mild-to-moderate distress patients experience following a transplant. For clarification both the title and the objective (in abstract) have been amended to reflect this.
P.3.	P.3. Strengths and limitations: "To reduce bias on the basis of the services offered to liver patients multiple sites with different organisation and delivery of services were selected." I realize the study was part of a larger project, but should this read as renal instead of liver patients?	Thank you for spotting this typographical error. Liver has been deleted and replaced with renal.
P.3.	P.3. "The diversity of participant characteristics supports the representativeness of the findings to this	 • We agreed this is not clear and have reworded this point. It now reads "Participants were diverse thus the findings

	patient group". What does this mean? Please explain.	are representative of the wider group of renal transplant patients".
P.5.	P.5. "Nevertheless, evidence suggests that distress, coping, and adjustment in transplant patients largely go undiagnosed or ignored and remain untreated. Currently, there is little evidence on mild-to-moderate distress in transplant patients and further research is required in order to understand the psychological and emotional effects of a transplant. " I agree, but there has been more published on distress and quality of life in transplant and specifically in the heart transplant literature. Perhaps that literature would augment this section and better situate the proposed article and its claims.	Agreed. To strengthen this argument, numerous references have been inserted into the manuscript.
P.5.	P.5. Methods: I found reading through the more detailed methods section in the published protocol useful in terms of recruitment (and refusals to participate).	Thank you for reading our protocol. Due to word count limitations we chose to include a brief summary of the methods and to direct the reader to the published protocol if they require more insight.
P.6.	P.6. The authors note that potential participants were contacted by one of the researchers. Which one? Was this person also directly involved in the patients' care? Would there be any patient concern about their care if they refused the researcher's request? How was this potential for more distress mitigated as part of the ethics process?	Thank you for raising these points. The initials of the contacting researcher should have been included in the manuscript and this has now been rectified (page 6). This person was an employee of Birmingham University and not involved in patient care. Participants indicated on the distress thermometer questionnaire whether they would be willing to take part in an interview and gave their preferred contact details. Therefore the participants knew about the study. Those expressing an interest were contacted to arrange an interview date after being sent a participant information sheet which outlined where distressed patients could get support, and assured patients that they could withdraw from the study at any time without their care being affected either before or after an interview. This is standard practice so has not been described in detail in the manuscript, but additional information on the recruitment process for the interviews has been provided on page 6.
P.6.	P.6. The authors note that both researchers identified as "female". Why was this language used instead of one centering gender (i.e. women-identified)? Was this important to the participants? How?	Noting the gender of the interviewers is good practice and recommended by the COREQ checklist. The wording in the document has been amended to say "both identified as female" (page 7). The gender of the interviewers was not important to the interviewees.

P.7.	It is important that the authors note the two interviewers did not previously know the participants. As per protocol around semi-structured interviews however, how was comfort and/or rapport built before the interviews and especially when conducted in the home? How much time was spent with participants prior to the formal interview? What kind of interviewing and/or qualitative research training do the interviewers/researchers have? Were there any differences in the depth and length of interviews between those interviewed in homes as opposed to over the phone?	Prior contact with participants (to answer queries about the study, organise the interview and making a reminder call) built up a rapport with the participants prior to their interview (page 6). The researchers are experienced qualitative researchers (page 7). There were no discernible differences in the depth and length of interviews between the two modes (page 21). All of these points are now included in the manuscript.
P.7.	Were participants shown how to turn off/control the recording if feeling uncomfortable or in distress?	Participants were not specifically shown how to turn off the recorder. However participants were advised to let the interviewer know if they needed to take a break during the interview or if they no longer wished to continue. This information has been added to the manuscript (page 7)
P.7.	With respect to the interview guide, was any opportunity given to participants to discuss other topics or areas of concern?	At the end of the interview each participant was asked if they wanted to add any other relevant information or whether they thought an important topic had been missed. The manuscript has been amended to reflect this (page 7).
P.7.	The authors note that "Analysis combined aspects of grounded theory (16) and thematic analysis (17). Interviews were initially analysed inductively using the open coding and constant comparison aspects of grounded theory. The initial coding framework was developed by JJ and CG and was appropriately refined following comparison and discussion. Transcripts were coded using NVivo 11." Why did the authors use grounded theory? What was the rationale? Where do the researchers position themselves ontologically and epistemologically vis a vis qualitative research? Were other qualitative methodologies considered and if so what and why? How was the decision made to move from 'thematic analysis' in the first published methods piece to grounded theory in this one? How do the researchers understand reflexivity and methodology? There needs to be more explanation and a clear rationale for theoretical and methodological decisions made by the research team for this project.	At the time of writing the protocol thematic analysis seemed to be the best approach. However, during the analysis of the qualitative data it became clear that there was no relevant theory or literature to draw upon so a hybrid approach was adopted.
P.8.	Table: Why equate white with British? How were these categories constructed and for what purpose? I found this section quite problematic.	Agreed British should not be equated to white, this has been removed from the table 2. The ethnicities reflect how the participants identified themselves on the distress thermometer survey, and the categories are taken from the English Office for National

		Statistics ethnicity groupings used in the national census. Using these categories is good practice for large-scale surveys.
P.16.	The authors note that “Many suggested that specialist psychological services should be available as in integral part of care for renal transplant patients.” Why just psychologists? Did participants discuss or were they made aware of other kinds of support professionals?	Discussion of relevant HCPs was dictated by those mentioned by the participants in the interviews. The interviewers did not suggest other professions to the participants. Given the nature of mild-to-moderate distress in renal transplant patients renal psychologists are perhaps the most appropriate health care professionals to help patients and are perhaps the HCPs that patients are more likely to know about, particularly as two of the participating NHS Trusts had psychologists on their renal staff. No changes have been made to the manuscript following this point.
P.17.	How many were surprised by the distress? In some themed areas, you list numbers of participants who spoke to the issue/theme. In others, you do not. Were the majority surprised and if so, please include	“Surprised” is perhaps the incorrect word to use here and has been changed to “largely unanticipated experience of distress” (Page 18).
P.17.	With respect to the gift, there is literature that identifies this as a major pressure for those with transplants. Please cite accordingly.	The reviewer is correct and appropriate papers are now cited (page 19).
P.18.	The authors note, “Given that patients receive little, if any, information about post-transplant distress from HCPs this is not surprising. Consequently, patients are often ill prepared and feel helpless when trying to cope with their distress.” Given this, it would appear that post renal transplant distress is not so unexpected, at least by the professionals who do not prepare them for it. Please revise this section accordingly.	We agree and welcome the opportunity to provide clarity and have revised this section (page 19/20).

Reviewer 2

Comment	Reply
I enjoyed reading this paper and felt that data was interesting and raised some important issues.	Thank you for reading our manuscript and we are pleased you found it interesting.
Summary: in the third point there is a reference to "liver patients", which I assume is a typographical error.	Apologies this is a typographical error and has been corrected.
Methods Table 1: Is it possible to briefly describe the basis for the classification of sites by size? Distress thermometer: Is there any information about sensitivity?	Thank you for this point. Sites were classified by size of their catchment area and this is now reflected in the manuscript (page 5). Detailed information about distress thermometers, including sensitivity, is available in the cited paper.

Results For example, I was struck by F413's account of stopping taking medication because of a felt lack of control. This didn't seem to me to fit very well with the sub-theme of the impact of medication and I wondered whether it could be integrated into the sub-theme of coping (coping and control having a close conceptual relationship). It would also be very interesting and, I think, important to understand precisely what sort of control the participant would have wished to have as this might be important for clinicians to understand given the seriousness of the consequences in this instance. I would suggest collapsing the sub-themes "Initial feelings following transplant" and "Association of distress to the transplant" to draw out more clearly what seems to be a central issue of the sorts of hopes and expectations that people have for the surgery and the apparent lack of preparation for the realities. Identifying and addressing unrealistic expectations in advance of surgery has already been identified as important in other areas of surgery, such as cosmetic and reconstructive procedures. In this respect it would be very helpful to see a quote to support the claim that "renal staff did not forewarn the participants" (P. 14/25). I would also suggest considering bringing the sub-theme "Lack of information about transplants and support" into this theme to pull together everything concerning expectations of surgery and provision of information and guidance prior to surgery. I think that this would also help to distinguish between two types of support that are currently combined but need, I would suggest, to be distinguished. There are support prior to surgery and post-surgical support.	This is an interesting point but we feel that this account is best placed in the medication sub-theme. The participant described stopping medication because of its side effects. We agree it would be interesting to understand what sort of control the participant wished to have but they did not describe this and it would be inappropriate for us to speculate. We agree with the reviewer that the sub-themes "initial feelings following transplant" and "association of distress to the transplant" should be collapsed into one sub-theme – "expectations of living with a transplant" (page 10) and we have collapsed these themes as suggested. Sub-theme "lack of information about transplants and support" is now under the theme "Why am I distressed" (page 10). A quote concerning renal staff not forewarning patients is now included in the manuscript (page 15).
Discussion I was not persuaded by the statement that the findings of this study do not support the report that fear of rejection declines with time. Although one participant spoke of fear of rejection it appears from what the authors report that the fear concerned the anticipated lifespan of the transplant rather than rejection per se.	Thank you for raising this issue. We appreciate that these are different things and we were referring to the lifespan rather than the rejection of a transplanted kidney. Wording in the manuscript has been amended accordingly (page 19).
NICE is "National Institute for Health and Care (not "Clinical") Excellence".	Thank you for spotting this error. It has been corrected (page 19).

VERSION 2 – REVIEW

REVIEWER	Jennifer Poole Ryerson University, Canada
REVIEW RETURNED	31-Dec-2019
GENERAL COMMENTS	Dear Authors/Researchers, Thank you for the opportunity to read your revised manuscript. I appreciate the work that went into responding to our comments and suggestions and the changes made to the document itself. I have four main areas of concern that are still outstanding.

	1. p.3. You changed the wording around the diversity of the sample, which is much improved, but I caution whether qualitative samples can ever be representative. I would suggest inserting the word 'more' to reflect this limitation and adding this to the limitations section at the end of the paper. 2. p. 7. You added in wording that demonstrated participants could stop the interview if they felt uncomfortable. Thank you. As some participants worry about the impact of this on their care, did you also let them know that this would not be the case? 3. p.8. In your responses to questions about methodology and analysis (in the Methods section), you noted, "At the time of writing the protocol thematic analysis seemed to be the best approach. However, during the analysis of the qualitative data it became clear that there was no relevant theory or literature to draw upon so a hybrid approach was adopted." I would argue that this is not the case, as the literature you have now added (i.e. Mauthner) provides both theoretical and methodological ways into qualitative inquiry on transplant and distress. In the revised paper, the analysis section thus needs more detail to reflect your decision-making process and why this literature was not used to make key analysis decisions. This is also a potential limitation of your study. 4. p.9. I am still confused by the conflation of sex and gender in the paper (and the limitation to just two choices). If this is how participants identified themselves when asked, please do clarify. Similarly, if participants identified their own ethnicity as 'white, Indian or Caribbean', please clarify that as well. If the research team decided on these categories and language, please explicate the rationale in the paper itself. Again, thank you for your work.
--	--

REVIEWER	Kenneth Gannon University of East London UK
REVIEW RETURNED	13-Jan-2020

GENERAL COMMENTS	Thank you for submitting the revised version of this very interesting paper. I think that the revisions have addressed the points that I raised in my review of the original submission
---

VERSION 2 – AUTHOR RESPONSE

Reviewer 1		
	Comment	Reply
P3	You changed the wording around the diversity of the sample, which is much improved, but I caution whether qualitative samples can	Thank you for this observation. We have updated the manuscript to include the word

	ever be representative. I would suggest inserting the word 'more' to reflect this limitation and adding this to the limitations section at the end of the paper.	"more" on page 3 and have included it in the limitations section.
P7	You added in wording that demonstrated participants could stop the interview if they felt uncomfortable. Thank you. As some participants worry about the impact of this on their care, did you also let them know that this would not be the case?	Participants were informed through the participant information sheet and prior to the start of the interview that taking part in an interview will not impact on their care. We have amended the manuscript to include this information.
P8	In your responses to questions about methodology and analysis (in the Methods section), you noted, "At the time of writing the protocol thematic analysis seemed to be the best approach. However, during the analysis of the qualitative data it became clear that there was no relevant theory or literature to draw upon so a hybrid approach was adopted." I would argue that this is not the case, as the literature you have now added (i.e. Mauthner) provides both theoretical and methodological ways into qualitative inquiry on transplant and distress. In the revised paper, the analysis section thus needs more detail to reflect your decision-making process and why this literature was not used to make key analysis decisions. This is also a potential limitation of your study.	Thank you for highlighting this oversight and prompting us to provide clarification on this issue. We have included in the manuscript the reason why we chose to adopt a pragmatic generic approach to analysis. Given that this was our initial aim we do not feel that this is a limitation to our approach.
P9	I am still confused by the conflation of sex and gender in the paper (and the limitation to just two choices). If this is how participants identified themselves when asked, please do clarify. Similarly, if participants identified their own ethnicity as 'white, Indian or Caribbean', please clarify that as well. If the research	Apologies for the confusion. We have replaced all instances of "gender" in the manuscript with "sex". All participants self-identified their ethnicity and their sex as male or female on the survey. Table 2 has been amended to show this information and a footnote added.

	team decided on these categories and language, please explicate the rationale in the paper itself.	
--	--	--

VERSION 3 – REVIEW

REVIEWER	Jennifer Poole Ryerson University, Canada
REVIEW RETURNED	06-Feb-2020

GENERAL COMMENTS	I thank the authors for all their careful revisions. I know only too well the effort necessary for a timely and fulsome response to reviewers' questions. The manuscript is much stronger for your edits, and much more clear in its intent, ethics and methods. I wish you all the very best with your research and most importantly, patient care.
--